# New Use for Old Drugs: The Protective Effect of Risperidone on Colorectal Cancer

**DOI:** 10.3390/cancers12061560

**Published:** 2020-06-12

**Authors:** Vincent Chin-Hung Chen, Yi-Hsuan Hsieh, Tzu-Chin Lin, Mong-Liang Lu, Yin-To Liao, Yao-Hsu Yang, Tsai-Ching Hsu, Robert Stewart, Jun-Cheng Weng, Min-Jing Lee, Wei-Che Chiu, Bor-Show Tzang

**Affiliations:** 1Department of Psychiatry, Chang Gung Memorial Hospital, Chiayi 61363, Taiwan; hjcch@cgmh.org.tw (V.C.-H.C.); jcweng@mail.cgu.edu.tw (J.-C.W.); geneelle@yahoo.com.tw (M.-J.L.); 2School of Medicine, Chang Gung University, Taoyuan 33302, Taiwan; 8802026@cgmh.org.tw; 3Department of Child Psychiatry, Chang Gung Memorial Hospital at Taoyuan, Taoyuan 33305, Taiwan; 4Bethel Psychiatric Clinic, Taipei 11074, Taiwan; jamaalbest@ym.edu.tw; 5Department of Psychiatry, Wan-Fang Hospital, Taipei Medical University, Taipei 11696, Taiwan; mongliang@tmu.edu.tw; 6Department of Psychiatry, School of Medicine, College of Medicine, Taipei Medical University, Taipei 11042, Taiwan; 7Department of Psychiatry, Chung Shan Medical University, Chung Shan Medical University Hospital, Taichung 40201, Taiwan; je2tezy@csmu.edu.tw; 8Health Information and Epidemiology Laboratory, Chang Gung Memorial Hospital, Chiayi 61363, Taiwan; r95841012@ntu.edu.tw; 9Department of Traditional Chinese Medicine, Chang Gung Memorial Hospital, Chiayi 61363, Taiwan; 10School of Traditional Chinese Medicine, College of Medicine, Chang Gung University, Taoyuan 33302, Taiwan; 11Clinical Laboratory, Chung Shan Medical University Hospital, Taichung 40201, Taiwan; htc@csmu.edu.tw; 12Institute of Biochemistry, Microbiology and Immunology, Chung Shan Medical University, Taichung 40201, Taiwan; 13Immunology Research Center, Chung Shan Medical University, Taichung 40201, Taiwan; 14Institute of Psychiatry, Psychology and Neuroscience, King’s College London, London WC2R 2LS, UK; robert.stewart@kcl.ac.uk; 15South London and Maudsley NHS Foundation Trust, London SE5 8AZ, UK; 16Department of Medical Imaging and Radiological Sciences, Chang Gung University, Taoyuan 33302, Taiwan; 17Department of Psychiatry, Cathay General Hospital, Taipei 10686, Taiwan; cgh05406@cgh.org.tw; 18School of Medicine, College of Medicine, Fu Jen Catholic University, New Taipei 24257, Taiwan; 19Department of Biochemistry, School of Medicine, Chung Shan Medical University, Taichung 40201, Taiwan

**Keywords:** antipsychotics, colorectal cancer, Taiwan National Health Insurance, risperidone, SW480

## Abstract

(1) Background: The potential of old drugs in novel indications is being greatly valued. We propose a triple-model study involving population-based, cell, and animal studies to investigate the effects of risperidone, a type of second-generation antipsychotic (SGA) drug, on colorectal cancer. (2) Methods: We used data from Taiwan’s National Health Insurance Research Database between 1997 and 2013 to compare 101,989 patients with colorectal cancer and 101,989 controls. Conditional logistic regression analyses were used to explore the association between SGA exposure and the risk of colorectal cancer. The following bench studies were performed to evaluate the findings of the population-based study. (3) Results: We found that SGAs had been less commonly used in colorectal cancer patients than in controls. The colorectal cancer risk was reduced with an increase in the cumulative defined daily dose (cDDD) of SGAs. The adjusted odds ratio of antipsychotic use for cDDD days was 0.32 (95% CI: 0.25–0.42). Risperidone exhibited the most prominent tumor inhibition effect in a cell screen study. Bench data revealed that risperidone significantly induced apoptosis and elevated intracellular ROS in human SW480 cells and suppressed the proliferation of the xenografted SW480 tumor in nude mice. (4) Conclusion: This triple-model study demonstrates the association between risperidone usage and a lower risk of colorectal cancer.

## 1. Introduction

Investigation of the treatment potential of old drugs in novel indications is currently encouraged through the application of existing health information data (e.g., National Health Insurance [NHI] records) because the long-term real-world data readily provides established effect and safety records [1] However, further validation is often required because of the limitations of observational data, particularly those from routine sources, including unmeasured confounding factors. For example, an epidemiology study of the potential protective effects of olanzapine on hepatic cell carcinoma could not be replicated from cell studies [2].

Researchers have been interested in the potential beneficial effects of antipsychotic agents on cancer risk because of the reduced risk of certain cancers, particularly prostate and colorectal cancer, reported in patients with schizophrenia [3,4,5,6] despite the substantial wider health disadvantages described in this population [7]. More specifically, some molecular studies have revealed that antipsychotics might have antiproliferative effects on tumor cells [8,9]. In a Danish population-based cohort study published in 2006, when compared with patients who had never used first-generation antipsychotics (FGAs), both women and men using FGAs exhibited a reduced risk of rectal cancer and women exhibited a reduced risk of colon cancer [10]. In the past decade, several new antipsychotics have been developed [11], and indications of these agents have been extended [12]. Antipsychotics are prescribed for patients with schizoaffective, bipolar, and major depressive disorders in addition to those with schizophrenia. However, no other population-based study has evaluated the risk of colorectal cancer among users of antipsychotics since the Danish study published in 2006, and associations with new generations of antipsychotics are unknown. We propose a triple-model study comprising a nationwide population-based case–control study, a cell study, and an animal study to assess the role of second-generation antipsychotics (SGAs) on the risk of colorectal cancer and identify the drug with the greatest potential.

## 2. Results

### 2.1. Population-Based Study

We extracted data for 101,989 patients with colorectal cancer and 101,989 matched controls for the period between 1 January 1997 and 31 December 2013. The demographics and clinical characteristics of the patients and controls are presented in Table 1. The association between the use of antipsychotics and the risk of colorectal cancer is summarized in Table 2. In the adjusted analysis, a higher cDDD is associated with a lower risk of colorectal cancer for all antipsychotics. The adjusted odds ratio (aOR) decreased as the cDDD increased. Among those with cDDD ≥168, the aOR was 0.32 (95% CI: 0.27–0.38). Trends of decreased risk among groups with higher cDDDs were observed among both FGA and SGA users (Table 3). The aORs for the ≥168 DDD group among FGA and SGA users were 0.36 (95% CI: 0.30–0.43) and 0.32 (95% CI: 0.25–0.42), respectively.

Further analyses of the associations between individual antipsychotics and colorectal cancer are presented in Table 3. For all antipsychotics, including clozapine, quetiapine, and risperidone, a higher cumulative dose was associated with a lower risk of colorectal cancer. In the sensitivity analysis, the inverse association was not changed to account for the influence of schizophrenia or bipolar disorder. The results of the sensitivity analysis of the negative control indicated that valproic acid usage was not associated with the risk of colorectal cancer.

### 2.2. Bench Studies

An MTT assay was first performed for clozapine, flupentixol, quetiapine, and risperidone to confirm their effects on the survival of SW480 cells. Notably, SW480 cells exhibited lower viability in the presence of 0.05, 0.1, and 0.2 mM of risperidone compared with the cells treated with other antipsychotics (clozapine, flupentixol, or quetiapine), respectively (Figure 1).

We conducted further cell and animal studies only for risperidone because risperidone exhibited the strongest toxic effect on the survival of SW480 cells. A normal colorectal epithelial cell line, CCD 841 CoN, was adopted as normal control. The results indicated that SW480 cells are more susceptible than CCD 841 CoN cells to the effects of risperidone after 48 h of treatment with risperidone than after treatment with other antipsychotics. Remarkable and dose-dependent cell viability was observed in SW480 cells that were treated with different concentrations of risperidone (Figure 2).

Moreover, considerably lower cellular viability was detected in SW480 cells than in CCD 841 CoN cells in the presence of 0.05, 0.1, and 0.2 mM risperidone (Figure 2). Accordingly, significantly increased sub-G1 population and Annexin V signaling were also detected in SW480 cells that had been treated with different concentrations of risperidone, and the effect was dose-dependent (Figure 3 and Figure 4).

To clarify the effects of risperidone on reactive oxygen species (ROS) generation, a dihydroethidium (DHE) assay was conducted. Considerably higher intracellular ROS level was detected in SW480 cells in the presence of 0.01, 0.05, 0.1, and 0.2 mM risperidone in a dose-dependent manner (Figure 5).

To further determine the effects of risperidone treatment in vivo, SW480 tumor xenograft experiments were performed. Notably, the mice that had been administered 1 mg/kg of risperidone had a markedly lower mean tumor volume than those in the control group toward the end of the experiments (Figure 6).

## 3. Discussion

To our knowledge, this is the first study to use a triple-model study design to investigate the association between SGAs and the risk of colorectal cancer. The major finding of our study is that SGAs were observed to be associated with a lower risk of colorectal cancer, as demonstrated by the population-based study. Risperidone, a type of SGA, exhibited the greatest potential tumor inhibiting effect in a screen cell study and could inhibit the proliferation of human colorectal adenocarcinoma cells in further cell studies and in the animal study.

The risk-lowering association of FGAs on colorectal cancer is consistent with the aforementioned Danish population-based study [10]. We extended the exploration of the possible risk-lowering association from FGAs to SGAs. As indicated in Table 3, FGAs and SGAs (such as clozapine, quetiapine, and risperidone) were associated with a lower risk of colorectal cancer. In addition, we conducted an analysis of the cumulative dose of antipsychotics to determine whether any dose-dependent effects existed. As displayed in Table 2, lower risk of colorectal cancer was associated with higher cumulative doses of both FGAs and SGAs.

Few studies have investigated the possible cancer-risk-reduction mechanism of SGAs. Notably, a study by Kline et al. reported that the administration of a selective antagonist of dopamine D2-like receptors, such as ONC201, disrupts dopamine receptor expression and activity, which leads to significant anticancer activity by activation of integrated stress responses [13]. The inhibition of the dopamine D2 receptor in pancreatic cancer cells reduced proliferation and migration and slowed the growth of xenograft tumors in mice [14]. Similar results were observed in the current study: risperidone markedly induced apoptosis in a colorectal adenocarcinoma cell line. Other carcinotoxic mechanisms have been explored in related studies.

Risperidone, at a dose of 0.25–1 mg/kg per day, inhibited 17-β-hydroxysteroid dehydrogenase 10 and inhibited prostate cancer growth by disrupting dihydrotestosterone synthesis in vivo in xenografts in mice [15]. In addition, an in vitro study reported that risperidone could increase the level of intracellular glutathione, which protects normal cells against oxidative damage [16]. Notably, mounting studies have emphasized the potential of modulating intracellular reactive oxygen species (ROS) levels for the development of cancer therapy [17,18]. Evidence has indicated that accumulation of ROS leads to photooxidative stress and cancer-specific cell death [19]. Moreover, antipsychotic drugs such as thioridazine have also been reported to cause apoptosis in various cancer cells by increasing the level of intracellular ROS [20]. Therefore, the findings in the current study suggest that ROS-mediated apoptosis could be a possible mechanism of action for killing colorectal cancer cells. Overall, the possible underlying mechanisms of the antitumor effect of risperidone are multifaceted and merits further investigations to clarify the precise mechanism of risperidone on colorectal cancer therapy.

## 4. Strengths and Limitations

To our knowledge, this is the first study that has used a nationally representative sample and a longitudinal data set to investigate the relationship between SGA agent exposure and the risk of colorectal cancer. The use of a national representative database implies that the findings are less susceptible to selection and recall bias. Several methodological limitations affect the inferences and interpretations of this population-based study. The level of adherence to prescribed medications was unknown, and the recorded prescription doses cannot be assumed to be equivalent to true exposure levels. In addition, several confounding factors were not available in the NHIRD, including lifestyle, smoking habits, and body mass index.

Regarding the xenografted animal model, another limitation of this study must be mentioned. A xenograft heterotopic model consisting of subcutaneous injection of tumor cells is a less expensive method that requires less time to establish and has better reproducibility [21]. However, it does not replicate the mechanisms involved in the development of human cancers. The features of the cancers, including growth, invasion, or metastasis, depend mainly on the interactions between xenografted tumor cells and the microenvironment of the host location [22]. Therefore, proper selection of grafting sites for tumor cells is crucial to replicate the mechanism of human cancer. For decades, orthotopic xenograft models of carcinoma have been recommended by many researchers because they provide a more similar microenvironment for tumors to develop and validate the treatment [23]. Although this study demonstrated the inhibitory effects of risperidone on the growth of xenografted SW480 cells through subcutaneous injection, orthotopic xenograft models using open surgical techniques, such as injecting tumor cells into the wall of distal rectum [24], are strongly recommended to retrieve more convincing data for understanding the precise mechanism of risperidone on inhibiting colorectal cancer.

The most profound strength of this study is that we created a triple-model study by using additional bench studies, including an in vitro (cell) study and an in vivo (animal) study, to validate the findings from population-based studies according to the aforementioned limitation. The results indicate a similar antitumor potential of risperidone on colorectal cancer. The use of different methods to validate the drug’s effects on cancer, in addition to using population-based studies, is recommended because of the aforementioned limitation. We observed that some findings from population-based studies cannot be replicated from cell studies, such as the effect of olanzapine on hepatic cell carcinoma [2]. In this study, we extended our previous dual-model study (population-based study and cell study) [2] to a triple-model study (population-based, cell study, and animal study) to detect the cancer-protective effect of specific drugs. The methods provide a research model, and real-world big data may become an essential source of information to discover new possibilities for anticancer drugs.

## 5. Material and Methods

### 5.1. Population-Based Study

The current study was approved by the Institutional Review Board of Chang Gung Memorial Hospital (approval number: 201700253B0C501). The samples were selected from the Taiwan National Health Insurance Research Database (NHIRD), under the aegis of the National Health Research Institute, which includes records of inpatient care, outpatient visits, dental care, medication prescriptions, medical procedures, and diagnosis coding [25]. The NHI program provides mandatory universal health insurance and covers the delivery of health care to 99.6% of the national population (i.e., approximately 22 million individuals). The data adopted in this study were obtained from the NHIRD for the period between 1 January 1997 and 31 December 2013.

Cases of colorectal cancer in this study were identified using International Classification of Diseases, Ninth Revision (ICD-9) codes 153 and 154, and these codes were required to be applied to at least three recorded outpatient visits within one year or to one admission diagnosis during the study period. In addition, the data were linked to the Catastrophic Illness Registry Dataset for reconfirmation of the diagnosis. In Taiwan, insured patients with colorectal cancer confirmed by pathology are eligible for inclusion on the Catastrophic Illness Registry and can apply for a catastrophic illness certificate. The issuance of this certificate requires confirmation of a diagnosis of a catastrophic illness by a physician and a formal review by the Bureau of NHI, which comprises a panel of medical experts. The index date for cases is the first date on which a claim related to colorectal cancer is made.

For each patient with colorectal cancer, we used an incidence density sampling method [26] and randomly selected one control without a colorectal cancer diagnosis before the case index date. The noncancer cohort comprised individuals who were selected from a data set of 1 million individuals without any type of cancer. Cases and controls were matched on a 1:1 ratio by age, sex, urban level of the region of residence, and insurance premiums. Individuals in the control group who had died or for whom health insurance coverage was discontinued before the index date were excluded.

The exposure of antipsychotic use among cases and controls was retrieved and quantified using the World Health Organization’s defined daily dose (DDD) [27]. To detect the dose effect, the cumulative DDD (cDDD) was graded as follows: 0–27, 28–83, 84–167, and ≥168 DDD. To avoid influence from protopathic bias, cases and controls with antipsychotic exposure in the year preceding the index date were excluded.

We listed all FGAs and SGAs and identified potential confounding factors. Commonly prescribed antipsychotics in Taiwan were classified as FGAs and SGAs for further analysis. FGAs comprised the following: chlorpromazine, levomepromazine, fluphenazine, perphenazine, prochlorperazine, triluoperazine, thioridazine, haloperidol, flupentixol, clopenthixol, chlorprothixene, zuclopenthixol, pimozide, loxapine, and sulpiride. SGAs comprised the following: ziprasidone, clozapine, olanzapine, quetiapine, amisulpride, risperidone, zotepine, aripiprazole, and paliperidone. Potential confounding factors were identified from the source data, including sex, age (at the index date), urbanization level, insurance premium, comorbidities, and concomitant exposure to medications. The comorbidities included depressive disorder, anxiety disorder, psychotic disorder, alcohol-related diseases, diabetes mellitus, hypertension, hyperlipidemia, chronic kidney disease, and chronic pulmonary obstructive disease. Concomitant exposure to medications covered aspirin, nonsteroidal anti-inflammatory drugs, and statins.

The demographic factors, comorbidities, and data on concomitant medication use were compared between the patients and controls. Demographic data included sex, age (at the index date), urbanization level, and insurance premium. Conditional logistic regression models were used to assess the association between different levels of antipsychotic exposure and the risk of colorectal cancer, and adjusted odds ratios (aORs) were calculated. To detect the confounding effect of diseases, we carried out further sensitivity analyses to adjust the influence of schizophrenia and bipolar disorder. We added valproic acid as the negative control to carry out another sensitivity analysis. All analyses were conducted using SAS version 9.4 (SAS Institute, Cary, NC, USA).

### 5.2. Cell Study

A colorectal adenocarcinoma cell line (SW480 [ATCC CCL-228]) and a normal colorectal epithelial cell line (CCD 841 CoN [ATCC CRL-1790]) were purchased from ATCC and were respectively cultured in Leibovitz’s L-15 medium or Eagle’s minimum essential medium. Based on the chemical structure categories, antipsychotics, namely, clozapine, flupentixol, quetiapine, and risperidone, were analyzed to explore their effects on cell survival. Cell survival was measured using the method described in our previous report [28]. The MTT assay was first performed for clozapine, flupentixol, quetiapine, and risperidone to confirm the drugs’ effects on the survival of SW480 cells. As risperidone exhibited the strongest toxic effect on the survival of SW480 cells, we conducted further cell and animal studies only for risperidone. Flow cytometry analysis and annexin V assay were performed. We provide detailed information of the cell study methods in the section of materials and methods.

#### 5.2.1. MTT Assay

A total of 5 × 10^3^ cells was seeded in each well of a 96-well plate, and cultured overnight at 37 °C. The culture medium was then replaced with a fresh medium containing different concentrations of antipsychotics. Triplicate treatments were conducted for each concentration. After 24 or 48 h of incubation, the culture medium was removed, 0.2 mL 3-(4,5-cimethylthiazol-2-yl)-2,5-diphenyl tetrazolium bromide (MTT) reagent (0.5 mg/mL) was added to each well, and the cells were incubated for another 4 h. Subsequently, 0.2 mL of dimethyl sulfoxide (DMSO) was added to each well of the plate to dissolve the formazan crystals, and absorbance was measured at 570 nm using a microplate reader (SpectraMax M5, Molecular Devices LLC, San Jose, CA, USA). The relative cell survival rate was calculated based on the ratio of absorbance of the treated sample relative to the absorbance of the control sample. As risperidone showed the most pronounced effect on tumor cell survival, we performed the following cell studies to specifically investigate the effect of risperidone.

#### 5.2.2. Flow Cytometry Analysis

For the flow cytometry analysis, cells were incubated with different concentrations of risperidone for 24 h. After incubation, cells were harvested, washed with phosphate buffered saline (PBS), and fixed with 70% alcohol for 12–16 h at 4 °C. The cells were then washed again with PBS and transferred into 12 75-mm tubes. A total of 10 μL of propidium iodide staining solution was added, mixed, and incubated in an ice bath in dark surroundings. After being filtered through a 40-μm nylon screen, the stained cells were analyzed using a FACSCanto II flow cytometer (BD Biosciences, San Jose, CA, USA).

#### 5.2.3. Annexin V Assay

To detect apoptosis, 1 × 10^6^ cells were treated with different concentrations of risperidone for 24 h. The cell pellets were collected after centrifugation and were resuspended in 100 μL of an annexin-binding buffer containing 1 μL of annexin V-fluorescein isothiocyanate (FITC). Next, 5 μL of annexin V-FITC and 1 μL of propidium iodide were added, and the samples were allowed to sit at room temperature for 15 min. The stained cells were immediately analyzed using a FACSCanto II flow cytometer (BD Biosciences).

#### 5.2.4. ROS Assay

Detection of reactive oxygen species (ROS) was conducted using a ROS detection kit (CAYMAN CHEMICAL, Ann Arbor, MI, USA). Briefly, SW480 cells were treated in different concentrations of risperidone for 16 h. According to manufacturer’s instructions, 5 µM DHE assay reagent was added and the cells were incubated at 37 °C for 30 min. After replacing the reaction solution with cell-based assay buffer, the fluorescence intensity was determined by FACS Calibur (Becton Dickinson, San Jose, CA, USA). N-acetyl cysteine and antimycin A were adopted as negative and positive controls, respectively.

#### 5.2.5. Statistical Analysis

Sample size was calculated using free sample size calculating software G*Power version 3.1.9.2 (Franz, Universitat Kiel, Kiel, Germany). With a power of 85%, 0.05 level of statistical significance, and effect size of 1.0, the sample size for each test was calculated to be 5. Fifteen mice were randomly assigned to one of the three experimental groups. SAS JMP 7.0 software (JMP, Cary, NC, USA) by one-way analysis of variance (one-way ANOVA), followed by Tukey’s multiple-comparisons test, was adopted to perform the statistical analyses. All values were shown as mean ± SEM. A *P*-value less than 0.05 was considered statistically significant.

### 5.3. Animal Study

A total of 15 male athymic nude mice (BALB/c nude mice) were used. All protocols were approved by the Institutional Animal Care and Use Committee of Chiayi Chang Gung Memorial Hospital, Taiwan (approval number: 2015040102). For the xenograft, SW480 cells (5 × 10^6^ cells in 100 μL of PBS) were injected subcutaneously into the flank of the mice when they were aged 6 weeks. We performed the animal study specifically for risperidone because we observed that risperidone had the strongest effect in a screen cell study. The doses of risperidone administered in the current study were in agreement with those in a study by Dilly et al. [15]. When the tumor volume reached almost 20 mm^3^, the mice were randomly divided into three groups as follows: the control group, the low-dose group, and the high-dose group. Every day, the mice were administered 0.25 mg/kg PBS or 1 mg/kg risperidone by oral gavage. Tumor diameters were measured weekly using a caliper, and tumor volumes were calculated. The mice were sacrificed after 5 weeks of treatment, and the tumors were harvested and weighed.

## 6. Conclusions

Conclusively, the epidemiological data showed that risperidone use was associated with a lowered risk of colon cancer, and in-vivo and in-vitro analysis confirmed the anticancer effect of risperidone in established cell lines.

## Figures and Tables

**Figure 1 cancers-12-01560-f001:**
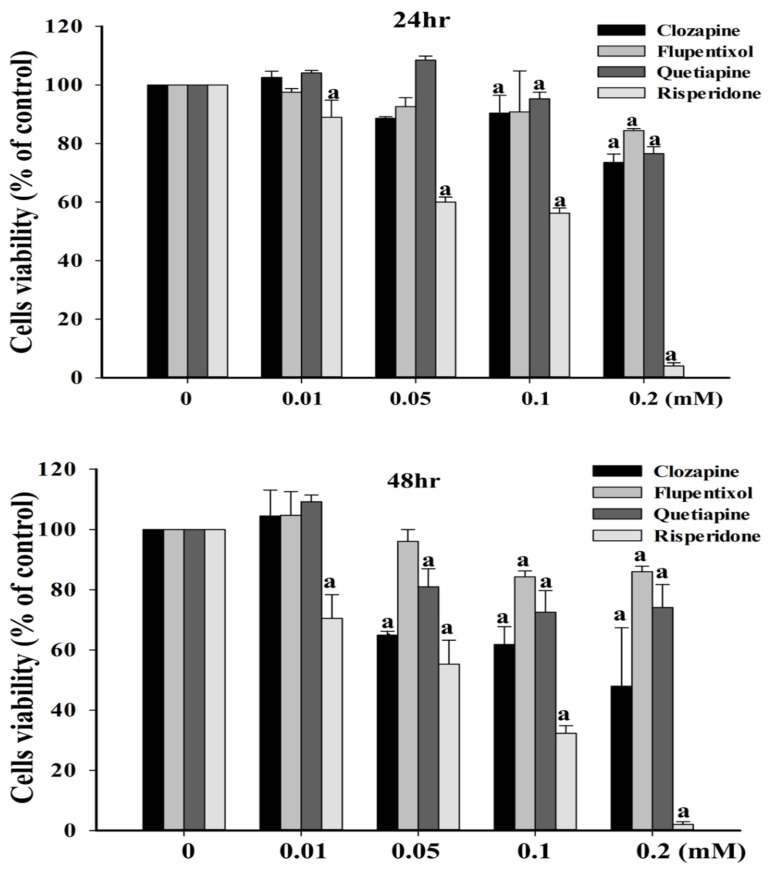
Effects of different antipsychotics on the survival of SW480 cells. “a” indicates a significant difference (*p* < 0.05) compared with clozapine, flupentixol, quetiapine, or the risperidone control (0 mM), respectively. Similar results were obtained in three to six repeated experiments.

**Figure 2 cancers-12-01560-f002:**
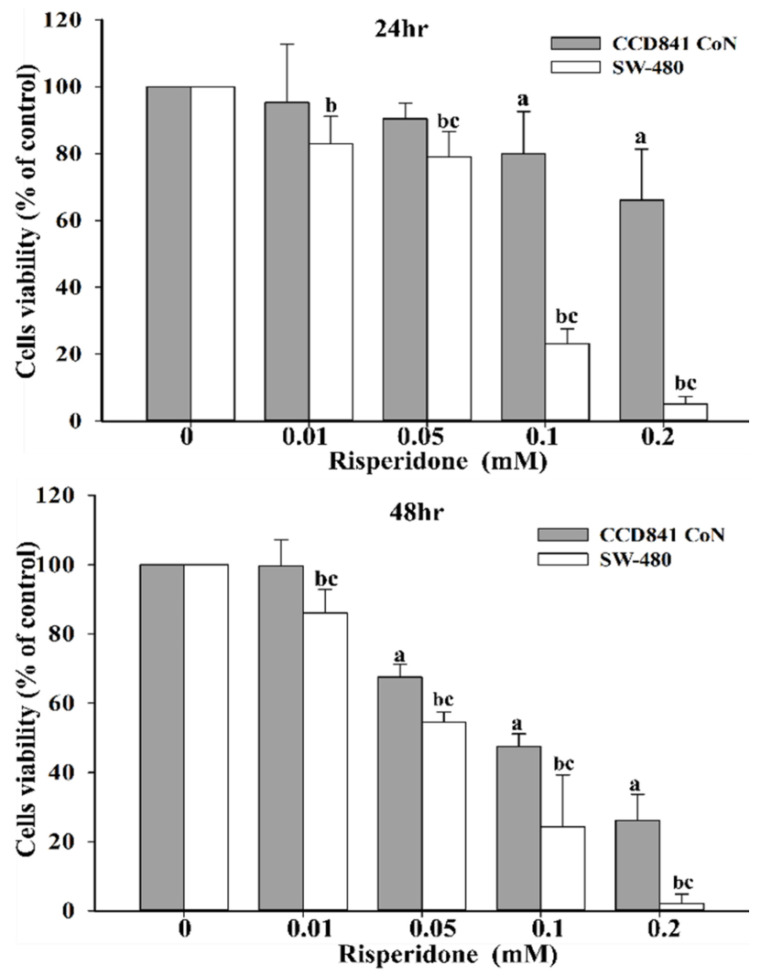
Survival of SW480 cells in the presence of risperidone. “a” indicates a significant difference compared with CCD 841CoN (0 mM). “b” indicates a significant difference compared with SW480 (0 mM). “c” indicates a significant difference between CCD 841CoN and SW480. The superscript letters “a,” “b,” and “c” indicate significant differences (*p* < 0.05). Similar results were obtained in three to six repeated experiments.

**Figure 3 cancers-12-01560-f003:**
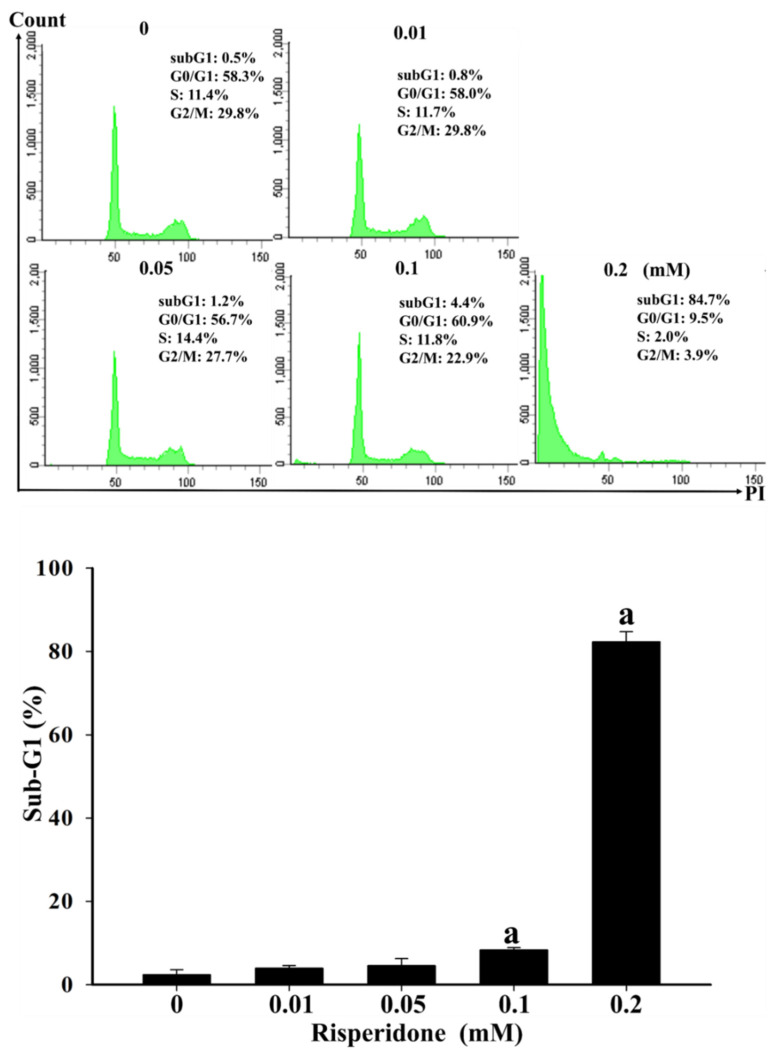
Sub-G1 proportion of SW480 cells in the presence of risperidone. “a” indicates a significant difference (*p* < 0.05) compared with the control group. Similar results were obtained in three to six repeated experiments.

**Figure 4 cancers-12-01560-f004:**
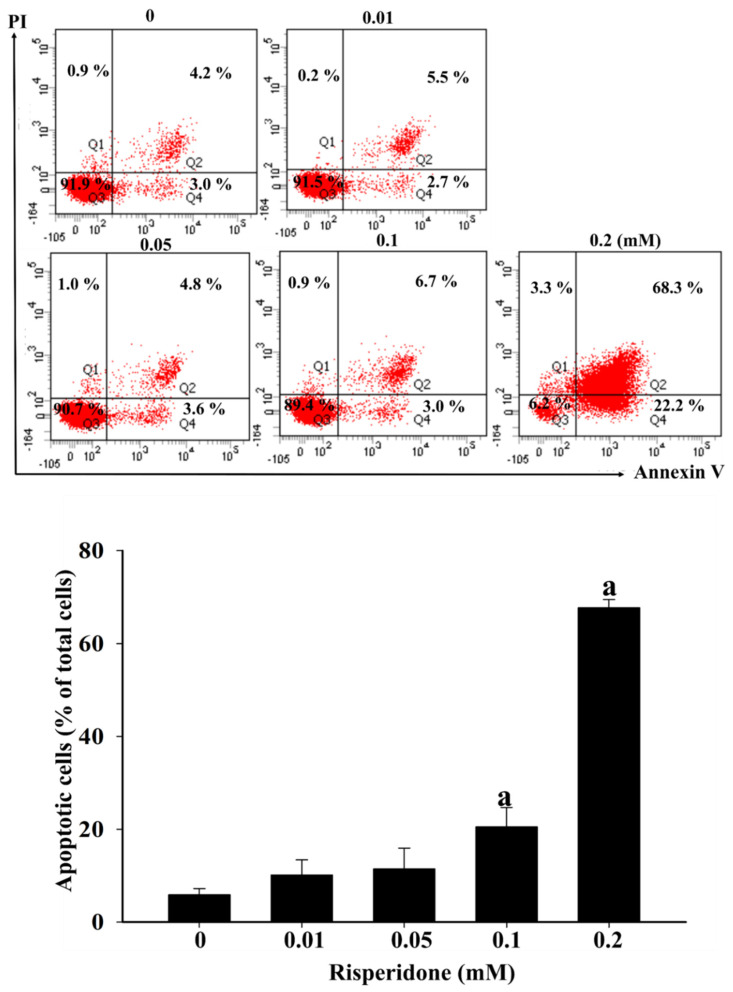
Apoptosis of SW480 cells in the presence of risperidone. “a” indicates a significant difference (*p* < 0.05) compared with the control group. Similar results were obtained in three repeated experiments.

**Figure 5 cancers-12-01560-f005:**
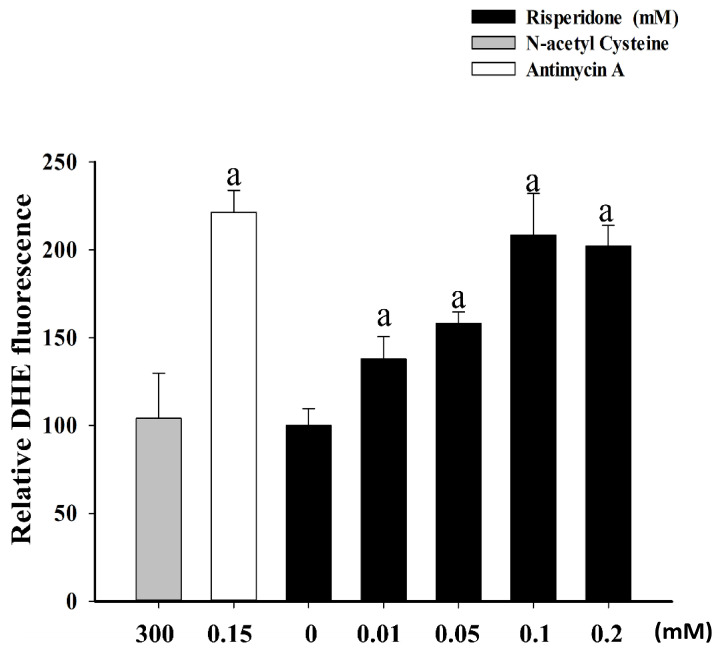
ROS concentration of SW480 cells in the presence of risperidone. “a” indicates a significant difference (*p <* 0.05) compared with the control group (0 mM). N-acetyl cysteine and antimycin A were used as negative and positive controls, respectively. Similar results were obtained in three repeated experiments.

**Figure 6 cancers-12-01560-f006:**
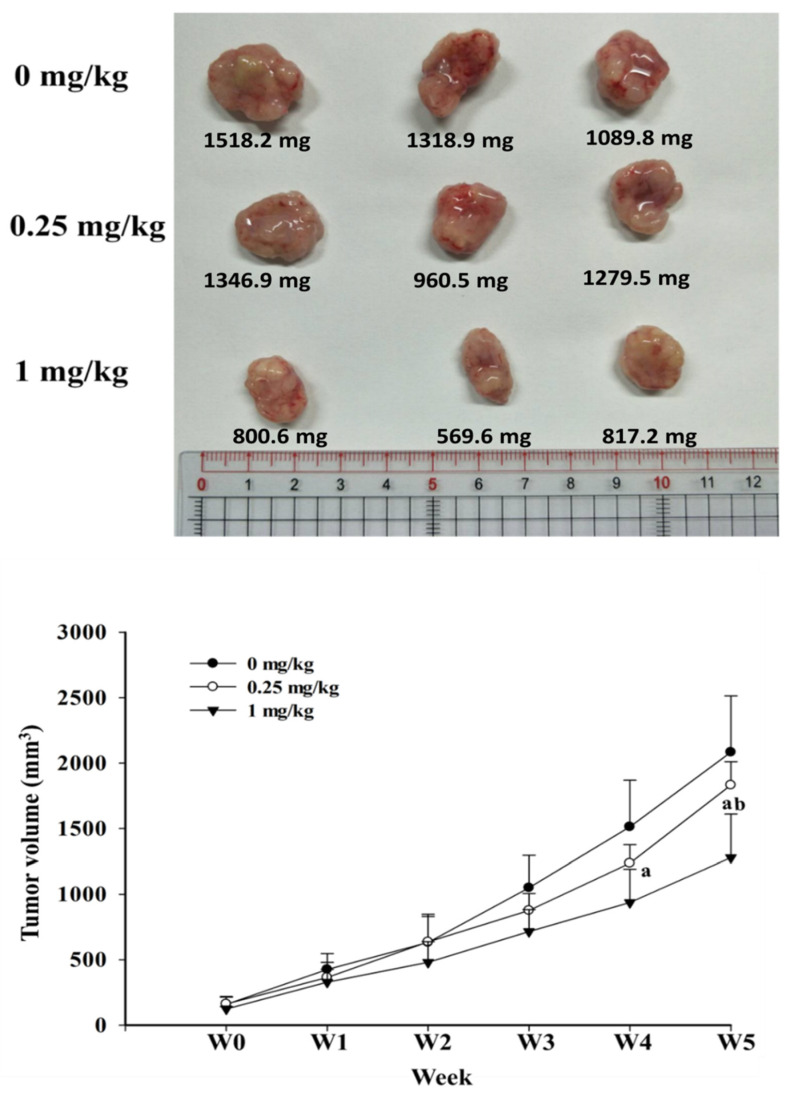
Effects of risperidone on xenografted SW480 cells. “a” indicates a significant difference compared with the control group. “b” indicates a significant difference compared with low-dose risperidone. The superscript letters “a” and “b” indicate significant differences (*p* < 0.05).

**Table 1 cancers-12-01560-t001:** Demographics and clinical characteristics of patients with colorectal cancer and those with noncolorectal cancer after exact matching in Taiwan between 1997 and 2013.

Characteristic	Colorectal Cancer, *n* = 101,989 (%)	Non-Colorectal Cancer, *n* = 101,989 (%)	*P*-Value
Gender			1.00
Female	49,357 (48.39)	49,357 (48.39)	
Male	52,632 (51.61)	52,632 (51.61)	
Age at index date, year			1.00
18–40	14,609 (14.32)	14,609 (14.32)	
40–50	24,788 (24.30)	24,788 (24.30)	
50–60	26,714 (26.19)	26,714 (26.19)	
≥60	35,878 (35.18)	35,878 (35.18)	
Residence			1.00
Low	7709 (7.56)	7709 (7.56)	
Moderate	16,871 (16.54)	16,871 (16.54)	
High	47,663 (46.73)	47,663 (46.73)	
Very high	29,746 (29.17)	29,746 (29.17)	
Insurance premium (NTD ^a^)			1.00
0	16,937 (16.61)	16,937 (16.61)	
1–25,000	15,559 (15.26)	15,559 (15.26)	
25,001–40,000	47,181 (46.26)	47,181 (46.26)	
≥40,001	22,312 (21.88)	22,312 (21.88)	
Aspirin, (>28 cDDD)	21,461 (21.04)	20,825 (20.42)	0.0005
NSAIDs, (>28 cDDD)	69,258 (67.91)	70,664 (69.29)	<0.0001
Statin, (>28 cDDD)	13,014 (12.76)	12,167 (11.93)	<0.0001
Antipsychotics, cDDD ^b^			<0.0001
0–27	100,535 (98.57)	99,647 (97.70)	
28–83	1016 (1.00)	1187 (1.16)	
84–167	251 (0.25)	389 (0.34)	
≥168	187 (0.18)	705 (0.69)	
FGAs ^c^, cDDD ^b^			<0.0001
0–27	100,580 (98.62)	99760 (97.81)	
28–83	973 (0.95)	1173 (1.15)	
84–167	246 (0.24)	351 (0.31)	
≥168	190 (0.19)	2313 (0.75)	
SGAs ^d^, cDDD ^b^			<0.0001
0–27	101,744 (99.76)	101,246 (99.27)	
28–83	134 (0.13)	272 (0.27)	
84–167	39 (0.04)	133 (0.13)	
≥168	72 (0.07)	338 (0.33)	
Medical diseases, yes			
Hypertension	36,970 (36.25)	34,465 (33.79)	<0.0001
Hyperlipidemia	19,176 (18.80)	18,467 (18.11)	<0.0001
Diabetes	19,349 (18.97)	16,031 (15.72)	<0.0001
COPD	16,080 (15.77)	15,845 (15.54)	0.15
Psychotic disorder	584 (0.57)	1386 (1.36)	<0.0001
Depressive disorder	3902 (3.83)	4733 (4.64)	<0.0001
Anxiety disorder	17,446 (17.11)	18,926 (18.56)	<0.0001
Chronic kidney disease	2662 (2.61)	1686 (1.65)	<0.0001
Alcohol-related disease	479 (0.47)	393 (0.39)	0.0035

^a^ USD = 32.1 New Taiwan Dollars (NTD) in 2008. ^b^ Drug dose usage presented in this table are cDDD one year prior to the index date. Abbreviations: NSAIDs, nonsteroidal anti-inflammatory drugs; COPD, chronic obstructive pulmonary disease; FGAs, first-generation antipsychotics; SGAs, second-generation antipsychotics. ^c^ FGAs include chlorpromazine, levomepromazine, fluphenazine, perphenazine, prochlorperazine, trifluoperazine, thioridazine, haloperidol, flupentixol, clopenthixol, chlorprothixene, zuclopenthixol, pimozide, loxapine, and sulpiride. ^d^ SGAs include ziprasidone, clozapine, olanzapine, quetiapine, amisulpride, risperidone, zotepine, aripiprazole, and paliperidone.

**Table 2 cancers-12-01560-t002:** Association between use of antipsychotics and risk of colorectal cancer in a population-based case–control study ^a^ in Taiwan between 1997 and 2013.

Variable	Unadjusted Analysis	Adjusted Analysis *
	Odds-Ratio (95% CI)	Odds-Ratio (95% CI)
Antipsychotics, cDDD ^b^		
0–27	1.00 (reference)	1.00 (reference)
28–83	0.85 (0.78–0.92)	0.88 (0.80–0.96)
84–167	0.64 (0.55–0.75)	0.70 (0.60–0.83)
≥168	0.24 (0.21–0.29)	0.32 (0.27–0.38)

* Adjusted for hypertension, diabetes, hypercholesterolemia, depressive disorders, alcohol liver disease, psychotic disorders, and use of aspirin, NSAIDs, or statin. ^a^ Exact matching by age, sex, region of residence, and insurance premium. ^b^ Drug doses used in this table are the cDDD one year prior to the index date. Abbreviations: CI, confidence interval; cDDD, cumulative defined daily dose.

**Table 3 cancers-12-01560-t003:** Analysis of the association between the use of individual antipsychotics and the risk of colorectal cancer in a population-based case–control study a in Taiwan between 1997 and 2013.

ATC-Code	Generic Name (cDDD ^a^)	colorectal Cancer, *n* = 101989 (%)	Non-Colorectal Cancer, *n* = 101989 (%)	Adjusted Odds-Ratio ^b^ (95% CI)
FGAs				
	0–27	100,580 (98.62)	99,760 (97.81)	1.00 (reference)
	28–83	973 (0.95)	1173 (1.15)	0.86 (0.78–0.93)
	84–167	246 (0.24)	351 (0.34)	0.77 (0.65–0.91)
	≥168	190 (0.19)	705 (0.75)	0.36 (0.30–0.43)
SGAs				
	0–27	101,744 (99.76)	101,246 (99.27)	1.00 (reference)
	28–83	134 (0.13)	272 (0.27)	0.57 (0.46–0.70)
	84–167	39 (0.04)	133 (0.13)	0.35 (0.24–0.51)
	≥168	72 (0.07)	338 (0.33)	0.32 (0.25–0.42)
N05AC02	Thioridazine			
	0–27	101,848 (99.86)	101,685 (99.70)	1.00 (reference)
	28–83	68 (0.07)	101 (0.10)	0.82 (0.60–1.13)
	84–167	14 (0.01)	49 (0.05)	0.35 (0.19–0.63)
	≥168	59 (0.06)	154 (0.15)	0.54 (0.40–0.74)
N05AD01	Haloperidol			
	0–27	101,859 (99.87)	101,555 (99.57)	1.00 (reference)
	28–83	76 (0.07)	179 (0.18)	0.53 (0.40–0.70)
	84–167	31 (0.03)	74 (0.07)	0.55 (0.36–0.85)
	≥168	23 (0.02)	181 (0.18)	0.20 (0.13–0.31)
N05AL01	Sulpiride			
	0–27	101,053 (99.08)	100,572 (98.61)	1.00 (reference)
	28–83	681 (0.67)	852 (0.84)	0.86 (0.78–0.96)
	84–167	171 (0.17)	247 (0.24)	0.78 (0.64–0.96)
	≥168	84 (0.08)	318 (0.31)	0.39 (0.30–0.49)
N05AH02	Clozapine			
	0–27	101,983 (99.99)	101,915 (99.93)	1.00 (reference)
	≥28	6 (0.01)	74 (0.07)	0.14 (0.06–0.33)
N05AH03	Olanzapine			
	0–27	101,967 (99.98)	101,864 (99.88)	1.00 (reference)
	28–83	5 (0.00)	31 (0.03)	0.22 (0.08–0.57)
	84–167	5 (0.00)	24 (0.02)	0.35 (0.13–0.94)
	≥168	12 (0.01)	70 (0.07)	0.32 (0.17–0.59)
N05AH04	Quetiapine			
	0–27	101,880 (99.89)	101,442 (99.46)	1.00 (reference)
	28–83	66 (0.06)	144 (0.14)	0.48 (0.36–0.64)
	84–167	19 (0.02)	93 (0.09)	0.22 (0.13–0.36)
	≥168	24 (0.02)	310 (0.30)	0.10 (0.06–0.15)
N05AL05	Amisulpride			
	0–27	101,981 (99.99)	101,926 (99.94)	1.00 (reference)
	28–83	6 (0.01)	18 (0.02)	0.48 (0.36–0.64)
	84–167	0 (0.00)	16 (0.02)	-
	≥168	2 (0.00)	29 (0.05)	0.14 (0.03–0.58)
N05AX08	Risperidone			
	0–27	101,874 (99.89)	101,587 (99.61)	1.00 (reference)
	28–83	69 (0.07)	151 (0.15)	0.59 (0.44–0.79)
	84–167	16 (0.02)	70 (0.07)	0.33 (0.19–0.57)
	≥168	30 (0.03)	181 (0.18)	0.27 (0.18–0.40)
N05AX12	Aripiprazole			
	0–27	101,979 (99.99)	101,957 (99.97)	1.00 (reference)
	28–83	3 (0.00)	13 (0.01)	0.34 (0.09–1.22)
	84–167	2 (0.00)	6 (0.01)	0.56 (0.11–2.88)
	≥168	5 (0.00)	13 (0.01)	0.72 (0.25–2.06)

^a^ Drug doses used in this table are the cDDD one year prior to the index date. ^b^ Adjusted for hypertension, diabetes, hypercholesterolemia, depressive disorders, alcohol liver disease, psychotic disorders, and use of aspirin, NSAIDs, or statin.

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
