# Peer review of "New Use for Old Drugs: The Protective Effect of Risperidone on Colorectal Cancer"

_cancers, 2020, doi:10.3390/cancers12061560_

Round 1

Reviewer 1 Report

The study “New use for old drugs: The protective effect of risperidone on colorectal cancer” by Chen et al have been revised and addressed all the comments. Therefore, this revised manuscript could be accepted by the journal in this present form.

Reviewer 2 Report

The manuscript is concisely written, and the research question is original. The methods were appropriate for the research question asked and the experiments were properly controlled. In my opinion, this study addresses a significant problem, and the aim and the application are achieved. I think that this report is acceptance.

This manuscript is a resubmission of an earlier submission. The following is a list of the peer review reports and author responses from that submission.

Round 1

Reviewer 1 Report

The study “New use for old drugs: The protective effect of risperidone on colorectal cancer” by Chen et al. is very interesting piece of work. In this work, the authors conducted a population-based study using an existing database and have successfully showed that second-generation antipsychotics (SGAs) drugs were associated with a lower risk of colorectal cancer. Authors further extended their study using in vitro and in vivo model and have also showed that treatment of risperidone (SGA) reduced in proliferation and inhibit tumor formation of human colorectal adenocarcinoma cells. Overall the manuscript is well written and this study reveals the potential to successful clinical application. Therefore this manuscript could be acceptable with some additional experiments that strengthen the study.

  1. Authors need to show (at least an experiment) the mechanism of action of risperidone on human colorectal adenocarcinoma cells.
  2. Details of statistical analysis (number of each experiment repeat, statistical inference etc.) and detail experimental procedures (flow cytometry, MTT assays) are needed to be incorporated in the material and method section.

Author Response

1. Answer:

Thanks for the valuable comment.

We have performed the ROS assay to investigate the possible action of risperidone on SW480 cells (Please see Fig. 1E in page 18-19).

In page 4, lines 17-18, we have added the following sentence,

“and elevated intracellular ROS in human SW480 cells”

In page 18, lines 5-8, we have added the following sentences,

To clarify the effects of risperidone on reactive oxygen species (ROS) generation, dihydroethidium (DHE) assay was conducted. Considerably higher intracellular ROS level was detected in SW480 cells in the presence of 0.01, 0.05, 0.1, and 0.2 mM risperidone in a dose-dependent manner (Figure 1E).

In page 19, lines 1-4, we have added the following sentence,

“Figure 1E. ROS concentration of SW480 cells in the presence of risperidone. “a” indicates a significant difference (P < 0.05) compared with the control group (0 mM). N-acetyl Cysteine and Antimycin A are used as negative and positive controls, respectively. Similar results were obtained in three repeated experiments.”

In page 22, lines 14-19 to page 23, lines 1-5, we have added the following sentences,

“Notably, mounting studies have emphasized the potentials of modulating intracellular reactive oxygen species (ROS) levels for the development of cancer therapy [17-18]. Evidence has indicated that accumulation of ROS leads to photooxidative stress and cancer-specific cell death [19]. Moreover, antipsychotic drug such as thioridazine has also been reported to cause apoptosis in various cancer cells by increasing the level of intracellular ROS [20]. Therefore, the findings in the current study suggest that ROS-mediated apoptosis could be a possible mechanism of action for killing colorectal cancer cells. Overall, the possible underlying mechanisms of the antitumor effect of risperidone are multifaceted and merits further investigations to clarify the precise mechanism of risperidone on colorectal cancer therapy.”

In page 35, lines 22-32, we have added the following references,

  1. Panieri, E.; Santoro, M. M. ROS homeostasis and metabolism: a dangerous liaison in cancer cells. Cell Death Dis. 2016, 7, e2253.
  2. Perillo, B.; Di Donato, M.; Pezone, A.;, Di Zazzo, E.; Giovannelli, P.; Galasso, G.; Castoria, G.; Migliaccio, A. ROS in cancer therapy: the bright side of the moon. Exp Mol Med. 2020, 52, 192-203.
  3. Buytaert, E.; Dewaele, M.; Agostinis, P. Molecular effectors of multiple cell death pathways initiated by photodynamic therapy. Biochim. Biophys. Acta Rev. Cancer. 2007, 1776, 86-107.
  4. Seervi, M.; Rani, A.; Sharma, A. K.; Santhosh Kumar, T. R. ROS mediated ER stress induces Bax-Bak dependent and independent apoptosis in response to Thioridazine. Biomed Pharmacother. 2018, 106, 200-209.

2. Answer:

Thanks for the valuable comment.

We have added the description of number of each experiment repeat in figure legends of Fig. 1A-1E.

In page 28, lines 18-19 to page 30, lines 1-18, we have added the description of experimental procedure, including MTT assay, Flow cytometry analysis, Annexin V assay, ROS assay, in the section of Materials and Methods.

In page 31, lines 2-5, we have also calculated the sample size and added the description in Materials and Methods section.

Sample size was calculated using free sample size calculating software G*Power version 3.1.9.2 (Franz, Universitat Kiel, Germany). With power of 80%, 0.05 level of statistical significance and effect size of 0.8, sample size for each test was calculated to be 8.

Reviewer 2 Report

The manuscript is concisely written, and the research question is original. The methods were appropriate for the research question asked and the experiments were properly controlled. In my opinion, this study addresses a significant problem, and the aim and the application are achieved. I think that this report is acceptance.

Author Response

Answer:

Thanks for suggesting acceptance of the manuscript.

Reviewer 3 Report

A very well designed and interesting study evaluating the use of anti-psychotics as a protective role in tumorigenesis. Authors used a trimodality approach - using epidemiological data from NHI database, then in vitro and in-vivo methods to evaluate efficacy/association with cancer. The authors' efforts are definitely commendable.

While the authors infer from the NHI database that patients on antipsychotics are at lower risk of developing colorectal cancer, it is based on retrospective/observational data, with its own caveats. In-vitro and in-vivo analysis shows effect only on established cancer cell lines.

A conclusion that antipsychotics are associated with lower risk of colorectal cancer can be misleading. It should be stated clearly that epidemiological data supports a lowered risk of colon cancer and in-vivo and in-vitro analysis confirms anti-cancer in established cell lines. However, authors shouldn't conclude from the in-vivo and in-vitro analysis that it decreases risk of cancer. Tumorigenesis is different from agent action on tumor.  

Author Response

Answer:

Thanks for the valuable comment.

As recommended by the reviewer, we have rewritten the following conclusion to avoid misleading.

In page 32, lines 8-10, we have added the following sentence,

“Conclusively, the epidemiological data showed risperidone use was associated with a lowered risk of colon cancer and in-vivo and in-vitro analysis confirmed anti-cancer effect of risperidone in established cell lines.”